# Heterogeneous Maturation of Arterio-Venous Fistulas and Loop-Shaped Venous Interposition Grafts: A Histological and 3D Flow Simulation Comparison

**DOI:** 10.3390/biomedicines10071508

**Published:** 2022-06-25

**Authors:** Balazs Szabo, Balazs Gasz, Laszlo Adam Fazekas, Adam Varga, Levente Kiss-Papai, Orsolya Matolay, Zsofia Rezsabek, Mohammad W. Al-Smadi, Norbert Nemeth

**Affiliations:** 1Department of Operative Techniques and Surgical Research, Faculty of Medicine, University of Debrecen, Moricz Zsigmond u. 22, H-4032 Debrecen, Hungary; szabo.balazs@med.unideb.hu (B.S.); fazekas.laszlo@med.unideb.hu (L.A.F.); varga.adam@med.unideb.hu (A.V.); rezsofi94@gmail.com (Z.R.); m.smadi@med.unideb.hu (M.W.A.-S.); 2Department of Surgical Research and Techniques, Faculty of Medicine, University of Pecs, Szigeti u. 12, H-7624 Pecs, Hungary; balazs.gasz@aok.pte.hu (B.G.); lev@eebt.hu (L.K.-P.); 3Department of Pathology, Faculty of Medicine, University of Debrecen, Nagyerdei krt. 98, H-4032 Debrecen, Hungary; orsolya.matolay@med.unideb.hu

**Keywords:** vascular graft, arterio-venous fistula, interposition venous graft, vascular regeneration, histology, intimal hyperplasia, flow characteristics, wall shear stress, 3D flow simulation

## Abstract

Vascular graft maturation is associated with blood flow characteristics, such as velocity, pressure, vorticity, and wall shear stress (WSS). Many studies examined these factors separately. We aimed to examine the remodeling of arterio-venous fistulas (AVFs) and loop-shaped venous interposition grafts, together with 3D flow simulation. Thirty male Wistar rats were randomly and equally divided into sham-operated, AVF, and loop-shaped venous graft (Loop) groups, using the femoral and superficial inferior epigastric vessels for anastomoses. Five weeks after surgery, the vessels were removed for histological evaluation, or plastic castings were made and scanned for 3D flow simulation. Remodeling of AVF and looped grafts was complete in 5 weeks. Histology showed heterogeneous morphology depending on the distribution of intraluminal pressure and WSS. In the Loop group, an asymmetrical WSS distribution coincided with the intima hyperplasia spots. The tunica media was enlarged only when both pressure and WSS were high. The 3D flow simulation correlated with the histological findings, identifying “hotspots” for intimal hyperplasia formation, suggesting a predictive value. These observations can be useful for microvascular research and for quality control in microsurgical training.

## 1. Introduction

In cardiovascular and vascular surgery, transplantation surgery, and reconstructive and plastic surgery, numberless vessel anastomoses and graft implantations are performed daily, including vascular access surgeries for hemodialysis procedures. Regeneration of the anastomoses, especially the maturation of fistulas and venous grafts, is strongly influenced by the hemodynamic forces driven by the flow characteristics, as supported by rich literature data [1,2,3,4,5,6]. Although numerous studies were published in the past decades, the exact mechanisms behind the anastomosis/fistula maturation and failure have not yet been fully elucidated.

Concerning the commonly used artificial arterio-venous fistulas and the rarely applicable loop-shaped grafts (arterial or venous interposition), failure is still a challenging problem [2,7,8]. Arterio-venous interposition grafts can be used in clinical reconstructive microsurgery when proper vessels are not available or long enough in free flap operations [9,10,11,12,13,14,15,16,17,18,19]. Loop-shaped arterial grafts can be used as an alternative method for vascular access when Cimino fistula creation is not possible [20,21,22]. However, little is known about the looped venous grafts. In this case, not only does the vessel geometry differ from the original condition, but so do the hemodynamic conditions, as happens with venous grafts in coronary bypass surgeries. The loop shape has a distinctive outer and inner side, which we suspect will correlate with a higher and a lower wall shear stress side, resulting from the flow pattern and Dean vortices. In this configuration, the pressure gradient is small; hence, the flow velocity throughout the graft is gradual.

In vivo, it is very difficult to directly measure all factors influencing the wall shear stress and related parameters, as well as the vascular geometry. In this sense, the rapidly developing flow simulation models provide useful estimation. Most of the vascular flow simulations are performed using computer-aided design (CAD) or 3D reconstructed models from CT and MRI images [23,24,25,26]. In CAD simulations, the vessel geometry is idealized, which does not fully represent the real condition. The 3D reconstructed models from MRI or CT images have a resolution constraint, as the vessels are reconstructed from 2D slices, causing loss of detail. In experimental conditions, an informative option is corrosion casting, where the vessels are filled with a polymer, and, after hardening, the tissues are dissolved away [27,28,29,30]. However, this technique does not allow a parallel histological examination and a comparison of the same vessel with a flow simulation that would allow a highly precise location of certain areas in vessel with specific blood-flow properties, even with prediction of histological changes.

In this experimental microsurgical study, our aim was to examine the regeneration and histological remodeling of microsurgically created arterio-venous fistulas and loop-shaped venous interposition grafts, together with 3D flow simulation analysis on special vascular castings to identify the coincidence of blood flow parameters which directly influence the maturation processes of grafts.

## 2. Materials and Methods

### 2.1. Experimental Animals

The experiment was registered and approved by the University of Debrecen Committee of Animal Welfare and by the National Food Chain Safety Office (registration Nr. 25/2016/UDCAW) in accordance with the national (Act XXVIII of 1998 on the protection and sparing of animals) and EU (Directive 2010/63/EU) regulations. Thirty male Wistar Crl:WI rats (8–10 weeks old, bodyweights: 349.7 ± 13.76 g) were randomly divided into sham-operated (Sham, *n* = 10), arterio-venous fistula (AVF, *n* = 10) and loop-shaped venous interposition graft (Loop, *n* = 10) groups. The rats were kept in standard cages, in alternating day and night light conditions in a 12 h cycle, with free access to drinking water and to conventional rodent chow.

### 2.2. Operative Techniques

Anesthesia was carried out via intraperitoneal injection of a mixture of ketamine (100 mg/kg), xylazine (10 mg/kg), and atropine (0.05 mg/kg). After the surgery, flunixin (10 mg/kg) was given subcutaneously for analgesia. Thrombosis prophylaxis was achieved with intravenous heparin (80 IU/kg) injection. The right inguinal region was shaved, disinfected, and then carefully isolated. An incision was made above the right inguinal ligament. The femoral artery and the femoral vein were gently dissected, as well as the second medial side branch from the inguinal ligament and the superficial inferior epigastric vein (Figure 1A).

Until this point, these steps were the same in every group. In the sham-operated group, after the vessel preparation, an additional 90 min passed before skin suture, to provide comparable duration with the operations of the other two groups. The skin was sutured with a 6/0 braided polyglycolic acid suture using a 3/8 cutting needle, applying horizontal mattress stitches.

In the AVF group, the distal end of the superficial inferior epigastric vein was cut and connected to the side of the femoral artery (Figure 1B). It was sutured with simple interrupted stitches, using 10/0 monofilament polyamide thread and a 3/8 tapered serosa needle. The fistula was then sutured to the muscle to keep it from moving and to avoid kinking of the vessel. Before closing the skin, the patency of the fistula was checked with the double occlusion test, also known as the “milking” test [31].

In the Loop group the femoral artery was clipped and cut through, and the removed section of the superficial inferior epigastric vein was inserted into the femoral artery, with two end-to-end anastomoses. Both anastomoses were sutured using simple interrupted stitches with the same 10/0 thread mentioned before. The loop shape was also secured using holding stitches just like in the AVF group (Figure 1C). If there was no bleeding, the skin was sutured as described before. The animals were observed for 5 weeks, with regular wound care. At the end of the follow-up period, the animals were anesthetized for the final examinations.

### 2.3. Histological Examinations

In the fifth postoperative week, under general anesthesia, using an operating microscope, the vessel samples were atraumatically dissected together with the surrounding muscle tissue, so that the geometry of the vessel remained intact. In half of the animals of the AVF group under the microscope, the vessels were cut in half at the middle of the anastomosis, as well as 1–2 mm after and before the anastomosis at the venous junction. The fistula was cut at three additional planes. Therefore, each location was examined in five different planes.

In half of the cases in the Loop group, the anastomoses were cut parallel to the vessel, and the loop itself was cut in three different planes. Before and after the loop, the vessels were cut perpendicularly. After the embedding of the samples in paraffin, the vessels were examined, and the important locations were marked for the comparison with the 3D flow simulation. Standard hematoxylin–eosin staining was used for histomorphological examinations.

### 2.4. Three-Dimensional Flow Simulation on Vessel Specimens

Vessel castings for 3D flow simulation were performed in the other half of cases in each group. The 3D CFD method was based on a protocol of lumen 3D scanning and application of standardized in silico examination of vessel specimens (ME3D-Graft, Hungary). Briefly, an ultralow-viscosity polymer was injected into the vessels, and then the tissue was removed from the hardened castings (Figure 2A,B).

Then, the lumen castings were scanned in high resolution for further applications using the simulation software with Ansys 20 R1 background (CARAT dental scanner, Kulzer, Japan, ME3D-Graft, Hungary). The scanner has a resolution up to 10 µm, which is smaller than a vascular endothelial cell [32,33]. The point cloud was converted to a 3D surface (stl) model and further modified to the igs file format of the NURBS (nonuniform rational B-spline) surface. The model underwent a standardized and protocol-based improvement for removing self-iterations, tubes, and holes using open-source software bundles (Meshmixer, Autodesk Inc, San Rafael, CA, USA, Geomagic Wrap, 3DSystems Inc., Valencia, CA, USA). The endings were elongated with at least three times the diameter; after simulation, the elongated parts of the models were not taken into consideration. A self-developed software was applied for computational fluid dynamics using Ansys 20 R1 background (Ansys, Inc; Canonsburg, PA, USA). Volume meshing was performed according to the protocol, resulting in high-quality mesh with at least 300,000 cells. This provided a standardized, scalable method for examination of microvascular structures.

A standard, transient fluid dynamic simulation was applied with preset conditions on the inflow and outflow of vessel structures. For boundary conditions of the average and normal rat vessel, the aortic velocity profile (distal aortic resistance and resistance of vena cava inferior) was applied. These were prerecorded in 100 timesteps per cardiac circle from invasive arterial pressure measurements.

More than 30 parameters were routinely measured, including spatial and time profile of velocity, pressure, vorticity, helicity, Reynold’s number, wall shear stress, and further derivates, such as oscillating shear index. In this study, we focused on the pressure and wall shear stress. For visualization and measurement of the co-appearance of pressure and wall parameter values, Ansys 20 R1 CFD-Post software was used (Figure 2C). Frozen frames from the simulation were used for the histological comparison. At definitive locations, the WSS and the pressure values were collected from the simulation model for numerical comparison and statistical analysis (Figure 3).

### 2.5. Statistical Analysis

To determine the sample size, Mead’s resource equation method was used. The statistical analyses for histological numerical data were performed using GraphPad Prism 8 software. All data distribution was checked for normality; accordingly, Student’s *t*-test, or Wilcoxon or Mann–Whitney nonparametric tests, as well as two-way ANOVA tests, were used. The significance level was set to *p* < 0.05.

## 3. Results

### 3.1. Geometry of the Fistulas and the Loops

Macroscopically analyzed vessel geometry data (from the photos taken with the microscope cameras just after the operation and in the fifth postoperative week) are summarized in Table 1.

In the AVF group, the outer diameter of the grafts in all measured locations were enlarged significantly by the end of the fifth postoperative week. This was seen only at the distal anastomosis in the Loop group. As shown by the alterations in the horizontal and vertical axis ratios, the shape of the fistula also changed (Table 1).

The graft became significantly shorter in the Loop group. The shape of the loops changed markedly, and the horizontal axes shortened significantly. The angle of the crossing between the axes of the distal and proximal ends of the looped grafts was 122.02° ± 5.39° just after the operation, whereas we recorded values of 71.80° ± 7.33° in the fifth postoperative week (*p* < 0.001).

### 3.2. Histomorphological Results

In the sham-operated group we found no changes. All of the arterio-venous fistulas matured during the follow-up period. In the Loop group, the grafts arterialized. The tunica media was enlarged significantly in both operated groups compared to the Sham vessels; however, the tunica intima was enlarged significantly only in the Loop (Figure 4).

In case of the AVF group, we examined the femoral artery proximally and distally to the anastomosis and at the anastomosis site, where the vein was also visible. We found a small patch of intimal hyperplasia opposite to the anastomosis orifice, as well as another one distally to the anastomosis in the artery. In the venous side of the anastomosis on the upper wall, we observed that the tunica media was significantly thickened compared to the lower wall. The lower wall only slightly enlarged compared to the intact vessel. Intimal hyperplasia formation was found in the fistula only at the junction of the superficial inferior epigastric vein and the femoral vein. It was located in the femoral vein where the blood rushed against the vein wall from the SIEV (Figure 5A). The diameter values were significantly enlarged at the distal region compared to the proximal end of the fistula (Figure 4D).

In the Loop group, the diameter became roughly the same throughout the graft after the arterialization. The tunica media was significantly thickened, almost in the entire area of the looped venous graft. Gradual enlargement toward the distal anastomosis was seen in the tunica intima layer, and the difference in thickness was significant at the two ends (Figure 4B). The outer and inner sides also showed asymmetrical intimal hyperplasia. The intima was much thinner in the area of the outer curvature at the distal section. At the proximal anastomosis, we did not find intimal hyperplasia formation; however, at the distal anastomosis, the tunica intima thickening was even on both sides, like the wall shear stress pattern (example in Figure 6).

### 3.3. Results of the 3D Flow Simulation of Vessel Specimens

Table 2 summarizes the wall shear stress (WSS (Pa)) and pressure values (mmHg) obtained from the 3D flow simulation tests. In the arterio-venous fistula, the wall shear stress was significantly decreased from the arterial to the venous direction, together with the pressure. WSS values were higher at the convex side of the curvature of the grafts. In the loop grafts, the pressure remained constant, while wall shear stress was moderately decreased from the proximal to the distal region of the venous interpositum graft (Table 2).

The simulations revealed the different flow pattern in the operated vessels in comparison to intact vessels. However, there were no alterations in the sham-operated vessel flow simulations. In case of the AVF, we also observed two “hotspots” of wall shear stress. One of them was found in the upper wall of the vein at the anastomosis site. Here, the wall shear stress showed much higher values at the upper venous wall compared to the lower. This was also seen in the arterial portion of the AVF (Table 2). The other “hotspots” were seen at the junction of the superficial inferior epigastric vein and the femoral vein. In this location, high wall shear stress was seen on the wall of the femoral vein opposite to the side branch opening (Figure 5A, Table 2). The pressure drop was much higher in the fistula than in the loop. In the first millimeters of the fistula, the pressure was higher at the arterial level, and the vein showed no dilatation after maturation.

In the matured fistula, approximately 2 mm away from the anastomosis, the vein suddenly dilated. Distally to this dilation point, the pressure and wall shear stress notably decreased. In the fresh fistula, the high wall shear stress was observed in a much longer section than in the matured one, but the pressure decreased gradually before the maturation.

In the loop, there was a clear difference in wall shear stress in the outer (convex) and the inner (concave) side of the curve, which became even again only at the distal third of the loop. The diameter of the original vein graft was slightly enlarged, which caused a decrease in wall shear stress in the distal part of the loop. Compared to the matured vessel, the geometry drastically changed. The diameter became even throughout the vein graft, and the size of the loop decreased (Table 1). This resulted in a homogenous distribution of wall shear stress at the peak of the pulse wave. There was no notable pressure drop between the two ends of the loop graft.

### 3.4. Comparison of Histological and 3D Flow Simulation Findings

The simulations showed homogenous wall shear stress and a gradual decrease in pressure in the sham-operated vessels. Accordingly, we found no changes in the histology either. At the anastomosis site of the arterio-venous fistula, the asymmetrical upper and lower wall shear stress shown by the simulation correlated with the asymmetrical tunica media enlargement of the vein. Where the pressure suddenly dropped, we found a major dilatation of the vein. The distal “hotspots” mentioned in the simulation coincided with the site of intimal hyperplasia found at the superficial inferior epigastric vein–femoral vein junction (see example in Figure 5). In case of the loop, we also observed a high correlation of the wall shear stress and the intimal hyperplasia. The outer and inner curvature intima pattern matched the asymmetric wall shear stress. Where the low wall shear stress became circular, we also found even intima enlargement at the distal anastomosis site (see example on Figure 6).

## 4. Discussion

It is known that, in the circulation, where the blood flow and the vessel wall are in constant interaction, the shear forces modulate the endothelial cells to produce various molecules and mediators (e.g., nitric oxide, prostacyclin, and thrombomodulin), while also orchestrating numerous other endothelial cell functions (ion channels, cellular signaling mechanisms, and transcription factors), as well as factors in vascular remodeling, via junctional mechanosensory complexes [34,35,36,37,38,39]. Flow characteristics in a new vascular geometry, therefore, affect the remodeling processes in anastomosis healing, fistula maturation, and arterialization of venous grafts, as well as intimal hyperplasia in atherosclerosis [2,40]. However, the exact role of shear stress in vascular wall remodeling has not yet been completely revealed. One of the theories claims that high wall shear stress causes intimal hyperplasia through various molecular pathways, including MVP-1 and MCP-2 [41,42]. Other experiments proved the opposite, i.e., low wall shear stress can be the cause [43,44]. Since intimal hyperplasia may occur in arterial bypass and arterio-venous fistula, where the conditions are different, it seems that shear stress is not the single factor responsible for intimal hyperplasia and other vessel wall alterations [40,45,46]. Certain combinations of shear stress, pressure gradient, pulsatility, vorticity, turbulent elements in flow characteristics, and blood rheology may induce these histological changes [1,2,35,45,47,48,49].

This experiment studied the combined effect of the wall shear stress and intraluminal pressure on the arterialization and fistula maturation processes by comparing the histological changes in the vessels with 3D flow simulation. The combined effect of these two parameters has not yet been thoroughly studied. The loop structure was applied as a model where the associations of different parameters could be examined. On that structure, the associations of multiple parameters formed demarked spots, raising the question of the effect of these spots on the physiology of maturation. These characteristics of parameter associations were further studied on morphologies of conventional AVF.

This method was modified as, instead of the conventional corrosion technique, the tissues were microsurgically removed. Our data showed that the flow pattern and the distribution of the wall shear stress strongly affected the maturation of arterio-venous fistulas and the arterialization of the loop-shaped venous interpositum grafts. An enlarged intima was found in both groups and in multiple areas, suggesting that different combinations of conditions can induce similar histological changes.

The histological analyses and the 3D flow simulations technically could not be performed on the same specimens. Therefore, half of the cases were sent for histology, while the remaining cases were used for casting, scanning, and 3D flow simulation. The fistulas and the loop grafts were performed by the same operator using the same technique, suture types, threads, etc. For accurate identification of the alignment of histological and CFD, we applied series of topographical markers and protocol-based cutting of specimens. The size of the vessels and the geometry of the fistulas and loops were also comparable, with relatively low standard deviation of the values (see Table 1 data just after operation).

Most of the histological changes we found are well documented in the literature, since arterialization and fistula maturation are known phenomena of vein grafts. During this process, the tunica media becomes enlarged, the vessel diameter can change, and the vein ultimately adapts to the new environment, through several different molecular pathways, including VEGF, TN-C, and ERK [42,50,51,52]. Intimal hyperplasia can be also formed, narrowing or even closing the vessel lumen. This is also orchestrated by many mediators, such as thromboxane, IGF-1, and TGF-B, as well as other factors [53,54]. All of these processes are affected by the blood flow properties, including pressure, flow velocity, and wall shear stress [35,44,47,49].

In the case of loop-shaped venous interpositum grafts, we observed the characteristic arterialization with tunica media enlargement without dilatation and intimal hyperplasia in some instances [48]. The superficial inferior epigastric vein also showed the typical signs of fistula maturation: enlarged diameter, moderate wall thickening, and possible intimal hyperplasia formation [54].

The arterialization processes homogenized the wall shear stress in the loop, correlating with the literature reporting that the morphological changes in an artery elevate shear stress, which can be an indirect sign of good perfusion [2,45]. The intimal hyperplasia location in the matured loop coincided with the low shear stress areas (the inner curve) shown by the simulation before the arterialization. This was probably due to the more homogenized flow pattern, showing the accuracy and the predictive capabilities of the used flow simulation techniques.

In the AVF group, the maturation behaved differently. Five weeks after the surgery, the flow pattern become more segmented. Interestingly, in the first millimeter of the vein, where the simulation revealed high wall shear stress and high pressure, a partial arterialization was found. The diameter remained the same, but the tunica media only thickened where both high wall shear stress and high pressure were presented at the upper venous wall. At the lower (posterior) wall, only moderate tunica media thickening was observable. This proves that, even though the pressure acts circumferentially, without the effect of wall shear stress, it does not cause tunica media thickening. The lower wall expressed lower shear stress, and intimal hyperplasia was not formed. We assume that the elevated flow velocity increased the wall shear stress above a currently unknown threshold. Therefore, even at the lower side, it was high enough to prevent the intimal hyperplasia formation. At the distal area, where pressure was moderate but wall shear stress was still uneven, no intimal hyperplasia formation was seen. It might be possible that the elevated pressure and wall shear stress prevented the intimal hyperplasia formation. Interestingly, when we examined the junction of the superficial inferior epigastric vein and the femoral vein, we found “hotspots” of high wall shear stress, as the arterial blood ejected from the superficial inferior epigastric vein and rubbed against the wall of the femoral vein. This is where intimal hyperplasia formation was also seen in a low-pressure environment. This suggested that both high and low wall shear stress can induce intimal hyperplasia formation depending on the relative ratio of pressure and wall shear stress.

As a limitation of the study, we used a rigid wall model during the CFD method. Although recent investigations suggested limited differences in fluid–structure interaction-based and rigid wall modeling regarding the presently observed parameters, the lack of calculation in the complex structure of loop and fistula formation might have influenced the exact location of parameter isometric surfaces. A further limitation of the present study is that we applied a marker-driven approach to identify CFD parameters and histological findings. Even though, in these specific samples and characteristic structures, we could achieve obvious identification and matching of the findings using these two modalities, we intend to apply a more accurate modeling tool and match 3D CFD data to 3D histological visualization for a more meticulous investigation in further, more detailed studies.

To summarize our findings, several observations can be highlighted. If both measured conditions (pressure and wall shear stress) are high, reflecting the arterial environment, arterialization may occur. However, without the high wall shear stress, high pressure can result in intimal hyperplasia formation. At a moderate level of pressure and wall shear stress, dilatation and tunica media thickening are the dominant alterations. If moderate pressure is presented at a low wall shear stress, the result may be varicosity [55]. If high wall shear stress is associated with low pressure, intimal hyperplasia may occur. Lastly, if both parameters are low, this reflects the venous environment.

The method introduced in this study can be an applicable tool not only for experimental studies but also for education of microvascular anastomosis techniques. The data on “hotspots” of arterialization, maturation, and intimal hyperplasia can provide an interactive method of microvascular anastomosis training to initiate best practices based on CFD observation. In one of our ongoing studies, the effectiveness of quality control for microvascular training is being examined. The ability to compare the performance of individual surgeons in microvascular suturing requires extensive study procedures and results [56]. The populational distribution of different CFD parameters with more than 1000 anastomoses is being investigated, enabling scoring and ranking of the suture-line performance. Comparative and regression-based studies are also being conducted to examine different CFD parameters in terms of experts’ opinions on microvascular anastomoses.

## 5. Conclusions

Regeneration and histological remodeling of microsurgically created arterio-venous fistulas and loop-shaped venous interposition grafts were completed in 5 weeks in rats. The maturation process distributed the pressure and wall shear stress distinctively and differently. The histological changes showed heterogeneous morphology depending on the distribution of intraluminal pressure and wall shear stress. The 3D flow simulation correlated well with the histological findings, identifying “hotspots” for intimal hyperplasia formation, suggesting a predictive value of this technique. The effect of coincident appearance of given pressure and WSS distribution can help further the understanding of mechanisms and outcomes of vascular anastomoses. These observations can be useful for further experimental vascular surgical/microsurgical studies for optimizing geometry of the fistulas and grafts, as well as for quality control in microsurgical training.

## Figures and Tables

**Figure 1 biomedicines-10-01508-f001:**
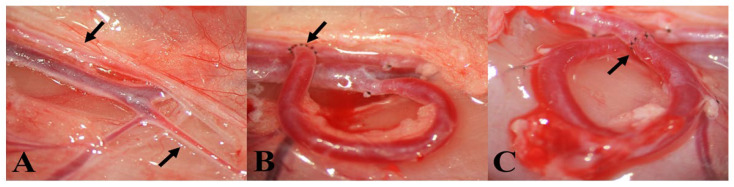
Photos of the vessels and the operation. (**A**) The dissected vessels. The top arrow shows the femoral artery, while the bottom arrow points to the superficial inferior epigastric vein (SIEV). (**B**) The finished arterio-venous fistula. The arrow shows the end-to-side anastomosis between the femoral artery and the SIEV. (**C**) The loop-shaped SIEV graft interpositioned into the femoral artery. The arrow points to the proximal anastomosis. Magnification: 10×.

**Figure 2 biomedicines-10-01508-f002:**
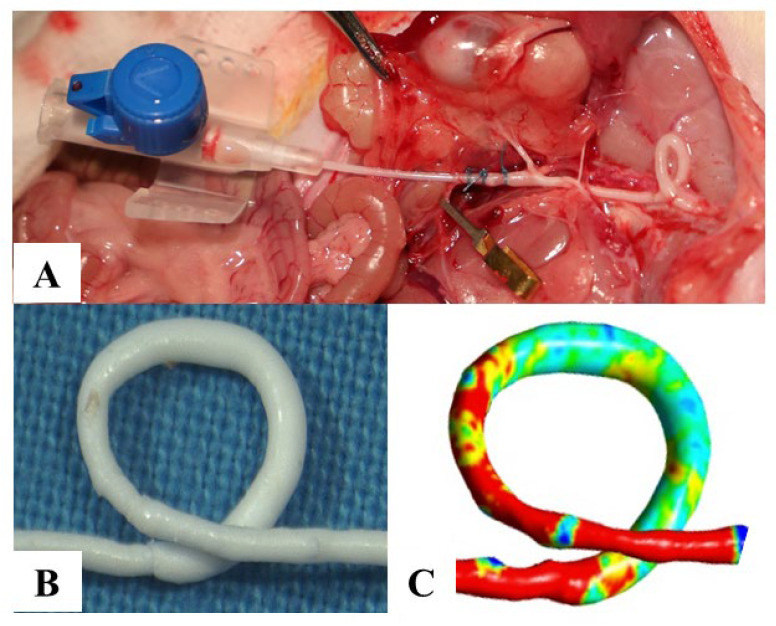
Main steps of the plastic casting process. (**A**) The graft was injected with the plastic from the right common iliac artery. (**B**) The plastic casting with the tissue removed. (**C**) The digitalized 3D vessel model. Magnification: 4×, 10×.

**Figure 3 biomedicines-10-01508-f003:**
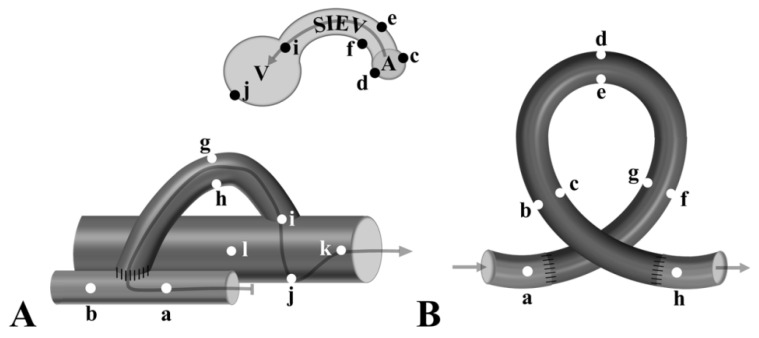
Schematic graphs of the AVF (**A**) and the loop (**B**) models showing the locations of the WSS and pressure measurements from the 3D flow simulation data.

**Figure 4 biomedicines-10-01508-f004:**
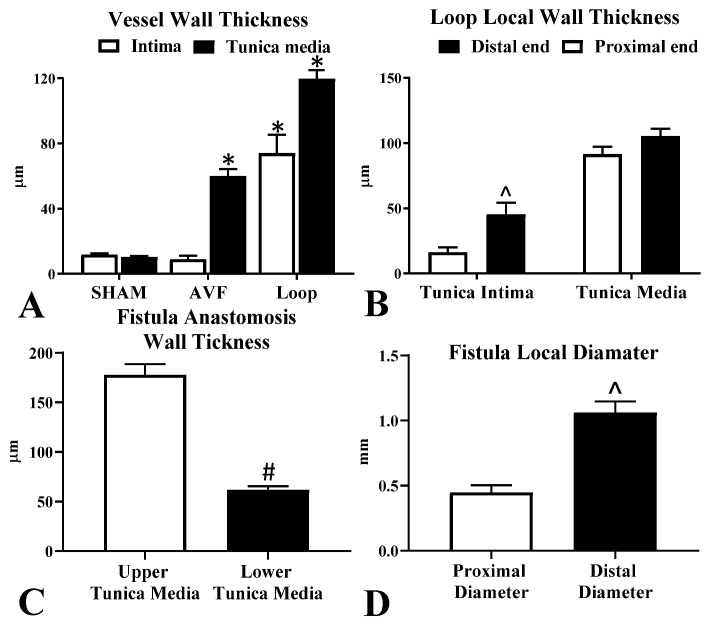
Numerical data of the histological results. (**A**) Comparison of the tissue layers (tunica intima and tunica media) between the groups. (**B**) Comparison of the tissue layers at the proximal and distal ends in the loop. (**C**) Comparison of the upper and lower walls at the AVF anastomosis site. (**D**) Diameter of the AVF at the proximal and distal area. Means ± SEM, *n* = 10; * *p* < 0.05 vs. Sham; ^ *p* < 0.05 vs. proximal end, # *p*< 0.05 vs. upper wall.

**Figure 5 biomedicines-10-01508-f005:**
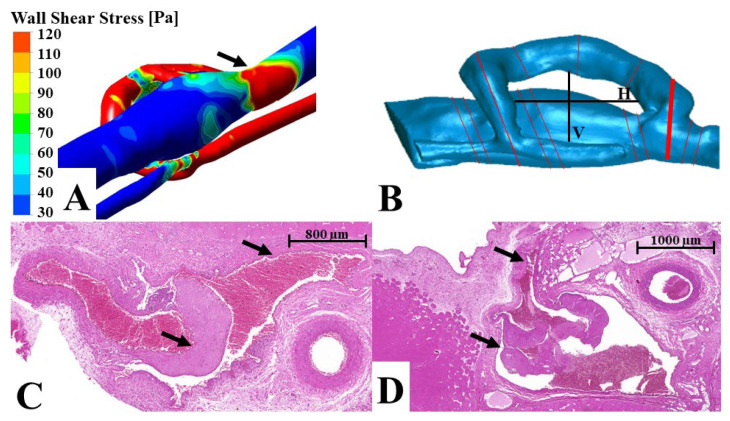
The junction site of the superficial inferior epigastric vein and the femoral vein—comparison of the flow simulation and the histology using a representative figure. (**A**) Flow simulation image of a fresh arterio-venous fistula. The arrow points to the vessel junction. (**B**) A 3D model of the vessels. The red lines show the plane where the tissue samples were sliced for histology. The black lines V (vertical) and H (horizontal) were used to evaluate the shape change of the fistula. The thick red line is represented in image (**C**). This is the venous junction of the superficial inferior epigastric vein and the femoral vein. (**C**,**D**) Histological slides of the superficial inferior epigastric vein–femoral vein junction. The upper arrow shows the superficial inferior epigastric vein, while the lower arrow shows the intimal hyperplasia formation in the femoral vein. Magnification: 50×.

**Figure 6 biomedicines-10-01508-f006:**
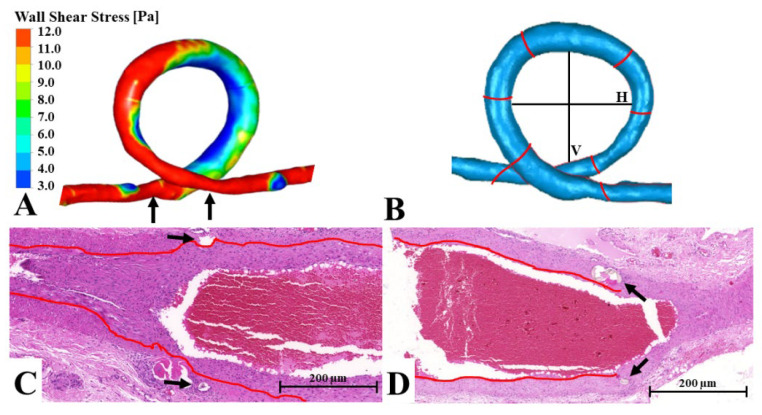
The loop—comparison of the flow simulation and the histology results using a representative figure. (**A**) Wall shear stress simulation of a freshly performed loop. The left arrow shows the distal anastomosis location, while the right arrow points to the area of the proximal anastomosis. (**B**) A 3D model of the loop. The red lines shows where the tissue samples were sliced for histology. The black lines V (vertical) and H (horizontal) were used to evaluate the shape change of the loop. (**C**,**D**) Longitudinal histological slide of the distal and the proximal anastomosis. The red line follows the internal elastic membrane. The arrows show the stitches at the arterio-venous junction. Magnification: 50×, 70×.

**Table 1 biomedicines-10-01508-t001:** Outer diameter (O.D.) values at various locations (mm), length (mm) and shape, as ratio (dimensionless) of vertical (V) and horizontal (H) axes of the loop-shaped fistulas or interpositioned grafts in the AVF and Loop groups just after the operation and in the fifth postoperative (p.o.) week.

Group	Time	O.D. at the Proximal Anastomosis	O.D. at the Curvature	O.D. at the Distal Anastomosis	Graft Length	H/V Loop Axis Ratio
AVF	Operation	0.99 ± 0.11	1.25 ± 0.25	1.12 ± 0.25	18.35 ± 2.5	0.74 ± 0.08
	5th p.o. week	1.40 ± 0.22	3.04 ± 0.71	2.67 ± 0.55	25.08 ± 3.21	1.16 ± 0.42
	*p*-Value	*<0.0001*	*<0.0001*	*<0.0001*	*<0.0001*	*0.006*
Loop	Operation	1.05 ± 0.08	1.38 ± 0.09	1.09 ± 0.08	19.04 ± 1.4	1.03 ± 0.09
	5th p.o. week	1.03 ± 0.1	1.37 ± 0.17	1.19 ± 0.09	16.13 ± 2.03	0.56 ± 0.15
	*p*-Value	0.759	0.921	*0.022*	*0.002*	*<0.0001*

Values are means ± SD.

**Table 2 biomedicines-10-01508-t002:** The 3D flow simulation data (wall shear stress (Pa) and pressure (mmHg)) at various sites of the loop-shaped fistulas or interpositioned grafts in the AVF and Loop groups. The 3D flow simulation was performed on the vessel molds taken and prepared in the fifth postoperative week.

Group	Localization	Site (Location in (Figure 3))	Wall Shear Stress (Pa)	Pressure (mmHg)
AVF	Artery, proximally to the anastomoses	(a)	24.61 ± 2.72 +	21.39 ± 2.92 +
	Artery, distally to the anastomoses	(b)	3.63 ± 0.49 *+	20.89 ± 3.23 +
	Arterio-venous anastomosis Arterial side	Upper (c)	40.74 ± 8.95 *+	17.01 ± 1.8 *+
		Lower (d)	4.16 ± 1.92 *+**×**	18.3 ± 1.85 *+
	Vein graft, proximal branch first mm	Upper (e)	41.04 ± 12.4 *+	15.5 ± 1.71 *+
		Lower (f)	6.23 ± 4.17 *+**×**	15.6 ± 2.03 *+
	Vein graft, middle (curvature)	Convex (g)	6.34 ± 1.02 *+	13.13 ± 1.19 *+
		Concave (h)	3.8 ± 0.55 *+#	13.06 ± 1.13 *+
	Venous junction (f)	SIEV (i)	0.28 ± 0.09 *+	12.33 ± 0.27 *
		Opposite (j)	2.28 ± 0.68 *+**×**	12.46 ± 0.56 *
	Vein, proximally to the anastomoses	(k)	0.49 ± 0.14 *	12.34 ± 0.22 *
	Vein, distally to the anastomoses	(l)	0.35 ± 0.13 *+	12.2 ± 0.12 *
Loop	Artery, proximally to the anastomoses	(a)	0.99 ± 0.13	57.89 ± 0.54
	Vein graft, proximal branch	Convex (b)	0.75 ± 0.27 *	57.98 ± 0.31
		Concave (c)	0.53 ± 0.06 *#	58.06 ± 0.26
	Vein graft, middle (curvature)	Convex (d)	0.23 ± 0.03 *	58.12 ± 0.21
		Concave (e)	0.18 ± 0.05 *#	58.12 ± 0.2
	Vein graft, distal branch	Convex (f)	0.3 ± 0.03 *	58.19 ± 0.13
		Concave (g)	0.31 ± 0.04 *	58.2 ± 0.15
	Artery, distally to the anastomoses	(h)	0.72 ± 0.17 *	58.22 ± 0.13

Values are means ± SD. * *p* < 0.05 vs. artery proximally to the anastomoses, # *p* < 0.05 vs. convex side of the bent graft, + *p* < 0.05 vs. vein proximally to the anastomoses, **×** *p* < 0.05 vs. SIEV/upper.

## Data Availability

The data presented in this study are available on request from the corresponding author. The data are not publicly available due to ethical constraints.

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
