# Peer review of "Heterogeneous Maturation of Arterio-Venous Fistulas and Loop-Shaped Venous Interposition Grafts: A Histological and 3D Flow Simulation Comparison"

_biomedicines, 2022, doi:10.3390/biomedicines10071508_

Round 1
Reviewer 1 Report
The manuscript entitled “Heterogeneous maturation of arterio-venous fistulas and 2 loop-shaped venous interposition grafts. A histological and 3D 3 flow simulation comparison” it is an interesting experimental manuscript. I have only few suggestion to improve manuscript. Please, see my specific comments below:
1. Was a power analysis performed to reach out the minimum number of subjects?
2. How about the study limitations? Could be interesting to include in discussion
3. Did the authors tried any multivariate analysis
Author Response
Dear Reviewer,
thank you very much for your time and for your valuable comments and positive opinion.
For the comments and questions we can provide the following responses:
- The Mead-equation was used when we planned the study, to determine the necessary case number.
- Concerning the limitations, we inserted the following text into the Discussion part: „As a limitation of the study, we used a rigid wall model during the CFD method. Although recent investigations suggest limited differences in fluid-structure interaction-based and rigid wall modelling regarding the presently observed parameters, the lack of calculation in the complex structure of loop and fistula formation might influence the exact place of parameter isometric surfaces. A further limitation of the present study is that we have applied a marker-driven approach to identify CFD parameters and histological findings. Even though in these specific samples and characteristic structures, we could find a way for obvious identification and matching the findings of this two modalities. It is intended to apply a more accurate modelling tool and match 3D CFD data to 3D histological visualization for more meticulous investigation serving the basis of further, detailed studies.
- Related to the limitations explained above, we have not apply multivariate analysis yet. In one of our ongoing studies on vascular regeneration, we will be able to set a better analysis approach, involving medical imaging data as well.
Thank you very much again for the valuable review. We hope that the answers and corrections in the revised version might be acceptable.
Sincerely yours,
Norbert Nemeth

Reviewer 2 Report
It is an excellent article. One aspect may be improved, they mentioned in the abstract :
“These observations can be useful for microvascular research and for quality control in microsurgical training as well.” That may be an important test for evaluating surgical skill but the authors did not describe how it can be done. They wrote few lines, discussion lines 405-408 and conclusion 419-421, on the subject.
Author Response
Dear Reviewer,
thank you very much for your time and for your valuable comments and positive opinion.
Concerning the skill assessment relation of the technique, we inserted the following text into the Discussion part: “In one of our ongoing studies, the effectiveness of quality control for microvascular training is examined. The ability to make comparisons of individual surgeons’ performance in microvascular suturing requires extensive study procedures and results [ref]. Populational distribution of different CFD parameters more than 1,000 anastomoses are being investigated, enabling scoring and ranking of the suture-line performance. The comparative and regression-based studies are further being conducted to examine different CFD parameters in terms of experts’ opinions on microvascular anastomoses.” And we have added a new reference here (Tolba et al. Eur. Surg. Res. 2017, 58, 246–262).
Thank you very much again for the valuable review. We hope that the answers and corrections in the revised version might be acceptable.
Sincerely yours,
Norbert Nemeth
